# Development of Amine-Functionalized Metal-Organic Frameworks Hollow Fiber Mixed Matrix Membranes for CO_2_ and CH_4_ Separation: A Review

**DOI:** 10.3390/polym14071408

**Published:** 2022-03-30

**Authors:** Naveen Sunder, Yeong Yin Fong, Mohamad Azmi Bustam, Nadia Hartini Suhaimi

**Affiliations:** 1Department of Chemical Engineering, Universiti Teknologi PETRONAS, Bandar Seri Iskandar 32610, Malaysia; naveen_20001953@utp.edu.my (N.S.); azmibustam@utp.edu.my (M.A.B.); nadia_17005943@utp.edu.my (N.H.S.); 2CO_2_ Research Center (CO_2_RES), R&D Building, Universiti Teknologi PETRONAS, Bandar Seri Iskandar 32610, Malaysia; 3Centre of Research in Ionic Liquids (CORIL), Universiti Teknologi PETRONAS, Bandar Seri Iskandar 32610, Malaysia

**Keywords:** metal-organic frameworks (MOFs), amine-functionalized, mixed matrix hollow fiber membranes, CO_2_/CH_4_ separation

## Abstract

CO_2_ separation from raw natural gas can be achieved through the use of the promising membrane-based technology. Polymeric membranes are a known method for separating CO_2_ but suffer from trade-offs between its permeability and selectivity. Therefore, through the use of mixed matrix membranes (MMMs) which utilizes inorganic or hybrid fillers such as metal-organic frameworks (MOFs) in polymeric matrix, the permeability and selectivity trade-off can be overcome and possibly surpass the Robeson Upper Bounds. In this study, various types of MOFs are explored in terms of its structure and properties such as thermal and chemical stability. Next, the use of amine and non-amine functionalized MOFs in MMMs development are compared in order to investigate the effects of amine functionalization on the membrane gas separation performance for flat sheet and hollow fiber configurations as reported in the literature. Moreover, the gas transport properties and various challenges faced by hollow fiber mixed matrix membranes (HFMMMs) are discussed. In addition, the utilization of amine functionalization MOF for mitigating the challenges faced is included. Finally, the future directions of amine-functionalized MOF HFMMMs are discussed for the fields of CO_2_ separation.

## 1. Introduction

Gas separation processes are widely used in industries for the separation of one or more gases from a mixture such as CO_2_/CH_4_ separation for natural gas sweetening and biogas purification [1], CO_2_/N_2_ separation from flue gas [2], and H_2_/CO_2_ purification in the water-gas shift reaction (WGSR) [3]. Several conventional gas separation technologies are available including adsorption [4], absorption [5], and cryogenics distillation [6]. However, this conventional technology faces some drawbacks, for example, the absorption method is less efficient with cost restriction, and faces difficulties in scale-up due to the complexity of the design process and cryogenic distillation high-energy requirement, high operation cost, and high tendency for blockage of process equipment [6]. Therefore, membrane-based technology has drawn more attention due to its energy efficiency, ease of scale-up, and environmentally friendly nature [7] compared with conventional methods.

Membrane separation technologies have been applied since 1980s in the natural gas industry to remove CO_2_, N_2_, H_2_S, and NGLs [8]. Membranes for gas separation are typically classified into three main groups defined by their materials of fabrication: polymeric, inorganic, and mixed matrix membranes. Still, trade-off between permeability and selectivity becomes an issue in polymeric membranes as indicated by the Robeson plot [9]. On the other hand, inorganic membranes having difficulties with regard to reproducibility as well as complicated fabrication methods [10]. To overcome these limitations and increase the performance of polymer membranes, inorganic materials are integrated into polymer matrix to form mixed matrix membranes (MMMs). Over the years, several types of conventional inorganic fillers were used in the development of mixed matrix membrane such as zeolites [3], carbon molecular sieve (CMS) [11], activated carbon [12], metal-organic frameworks [13,14,15,16,17,18,19,20,21], carbon nanotubes (CNTs) [22], metal oxides, mesoporous materials, non-porous material, and lamellar inorganic materials [23]. Common types of MMMs contains two phases which are the polymer (continuous) phase and a dispersed inorganic phase [23]. Incorporation of inorganic filler in the polymer matrix should provide better separation performance due to their excellent chemical and thermal stability, higher affinity to CO_2_ molecules and molecular sieving effect [24]. Moreover, organic polymers are easily subjected to physical aging due to the imbalance in its structure which reduces their fractional free volume and that leads to an increase on the membrane’s gas permeability [25]. Despite that, with sufficient filler loading and good compatibility, the physical aging resistance of MMMs can be improved because the increased binding is forced between the filler’s surface and the polymer matrix as the MMM becomes more rigid and less prone to collapse [26].

Metal-organic frameworks (MOFs) are a satisfactory alternative to improve the permeation properties of membranes because of its large surface area, high porosity, adsorption capacity, and good compatibility [15]. This has led to the utilization of MOFs in numerous applications which include gas/liquid separation, catalysts, medicine, and electronics [27]. However, pure MOF film membranes are highly brittle and difficult to synthesize [28]. Consequently, MOFs based mixed matrix membranes have gained attention in the past few years as an alternative to improve on pure MOF film membranes. Yet, filler particle aggregation and interfacial defects are the major drawbacks that hinder the development of MOFs filler based MMMs. Due to accessible metals and ligands used in MOFs synthesis, modified structures can be tailored to enhance the interaction between MOF filler and polymer chains as well as provide exceptional physical and chemical properties [15]. Therefore, MOFs have become part of the selection of fillers for the development of MMMs for optimizing gas diffusion and selectivity.

The development of amine-functionalized MOFs mixed matrix membrane and hollow fiber mixed matrix membrane is potentially to be explored in gas separation. Incorporation of amine-functionalized MOFs into a polymer matrix are among the current efforts aimed to provide enhanced performance for CO_2_ gas separation. Therefore, this review aims to addresses recent trends in the use of amine-functionalized MOF in the fabrication of mixed matrix membrane for the improvement of gas separation performance of membranes, mainly on hollow fiber configuration. The challenges in the development of amine-functionalized MOFs based mixed matrix membranes for flat sheet and hollow fiber configurations will be highlighted.

## 2. Separation Mechanisms of Hollow Fiber Mixed Matrix Membranes

Gases passage through amine-functionalized MOF based hollow fiber mixed matrix membranes (HFMMMs) can incorporate several possible mechanisms such as solution diffusion [29], molecular sieving [30], and Knudsen diffusion [31,32] as shown in Figure 1.

The solution diffusion mechanism works in accordance with the solubility and diffusivity of the gas molecules in the polymer of the HFMMM. The gas molecule is initially absorbed into the polymer and diffusion in the polymer happens from a higher pressure or chemical gradient (upstream phase) and towards the lower gradient (downstream phase) before desorption from the polymer occurs [29]. Notably, the solution diffusion mechanism is the main mode of transportation for gas molecules. The separation process through the dense layer is quantified via permeability coefficient, P (cm^3^(STP)·cm/cm^2^·s·cmHg) measured in Barrer for flat sheet membranes, and permeance (cm^3^(STP)/cm^2^·s·cmHg) measured in GPU for hollow fiber configuration, a product between diffusivity coefficient, D (cm^2^/s), and solubility coefficient (cm^3^(STP)/cm^3^·cmHg). Undesirable interfacial defects may form surrounding the metal-organic framework (MOF) in the MMM due to the incompatibility between the MOF and the polymer [31] or surface delamination of the polymer chain during the composite membrane fabrication [26]. This can lead to a decrease in performance namely the gas pair selectivity of the membranes as the nanometer interfacial voids will allow larger gas molecules such as CH_4_ to easily pass through the membrane surface. This type of transport mechanism is known as Knudsen diffusion as shown in Figure 1. Amine-functionalized MOFs generally have an excellent interaction with polymer matrices therefore minimizing macrovoid formation on the membrane surface [33]. Molecular sieving is another method for gas transport to occur in MOFs whereby the gas molecules are separated based on the kinematic diameter of the gas molecule and the aperture size of the MOF particle matrix. The inability for larger kinetic diameter gases to pass through the smaller apertures of MOFs results in a molecular sieving being a highly selective mechanism for gas separation [30]. Yet, the study on the gas transport mechanism towards HFMMMs is still lacking and requires further research in order to accurately determine the various methods by which the gas molecules diffuse through the HFMMM layer compared with flat sheet configured membranes as well as how amine functionalization may enable better transport properties in HFMMMs.

## 3. Metal-Organic Framework Fillers

Metal-organic frameworks (MOFs) are porous materials constructed by using metal ions (or clusters) and organic linkers, which consolidate the features of both organic and inorganic components. There are a variety of MOFs constructed from diverse materials, and UiO-66 [34,35,36,37], MIL-53 [3,16,38,39], and ZIFs [40,41,42,43,44] are the type of MOFs which are commonly reported in the literature due to their exceptional properties which includes high porosity [45], large surface area, and variable pore size and apertures [46], fine tunable chemistry [25], and good thermo-chemical stability [47,48]. MOFs can also be synthesized in different sizes and dimensions from 0-D to three-dimensional by manipulating the ligands additives used and the reaction conditions [27].

Zr-based MOFs such as UiO-66 and UiO-67 (University of Oslo (UiO)) is created by oxo cluster nodes and linear dicarboxylate linkers [49]. UiO-66 is a zirconium-based MOF with the formula of Zr_6_O_4_(OH)_4_(-BDC)_6_, with a Langmuir surface area of 1187 m^2^/g, and thin triangular windows with an aperture size of 6.2 Ȧ [36,50]. Due to the zirconium metal center being connected to the BDC-linker (benzene-1,4-dicarboxylate), it allows UiO-66 to achieve a superior chemical and thermal stability. Meanwhile, MOF UiO-67(Zr) contains zirconium ions and biphenyl-4,4′-dicarboxylic acid. UiO-67 has an outstanding chemical stability, relatively large specific surface area, lower density, and microspore volume, with two types of cages including super-tetrahedral cages (11.5 Å) and super-octahedral cages (18.0 Å), with accessible microporous triangular window openings (8.0 Å) [51].

Materials Institute Lavoisier (MIL) is a type of MOF that is developed by connecting metal ions such as aluminum, chromium, and titanium, to organic linkers. MILs have been categorized on the basis on their distinguishable porous structure, which consists of different metallic ion clusters intimately connected with specific organic carboxylates, as well as their separate pore system and chemical composition, whilst numbers are names of MILs that represent specific structures with different metal ions and oxidation states [52]. Besides that, the aperture size of most MIL MOFs is larger than the kinetic diameter of H_2_ (2.9 Å), CO_2_ (3.3 Å), N_2_ (3.6 Å), and CH_4_ (3.8 Å) gas molecules [26,53].

MIL-53 (Al) is created by connecting aluminum-based octahedra AlO_4_(OH)_2_ with the 1,4-benzenedicarboxylate (BDC) ligand and has a chemical composition of Al(OH)(O_2_C-C_6_H_4_-CO_2_) [39,52]. MIL-53 is a three-dimensional porous structure with one-dimensional diamond-shaped channels [25,54]. MIL-101 (Cr) is composed of octahedral clusters of trimeric chromium (III) interconnected by 1,4-benzenedicarboxylates with the chemical formula of Cr_3_F(H_2_O)_2_OO_2_C-C_6_H_4_-CO_23_·nH_2_O [52,55]. On the other hand, a newly discovered titanium-based MIL-125 (Ti) MOF consists of 2 cage sizes of 6.1 Å and 12.6 Å with pore apertures of 6 Å, also possibly a smaller aperture when amine functionalization is introduced. The three-dimensional configuration of MIL-125 has octa-nuclear Ti-clusters which allows octahedral vacancies thus forming a quasi-cubic tetragonal structure with a chemical formula of (Ti_8_O_8_(OH)_4_[O_2_C-C_6_H_4_-CO_2_]_6_) [17,56].

Zeolitic imidazolate frameworks (ZIFs) are derived from zeolites but with tunable pore structures and higher thermal and chemical stability, therefore making them excellent sieving MOFs [42,43,57]. Common ZIF materials such as ZIF-7 and ZIF-8 are produced by varying the organic linkers of imidazolate ligands along with the metal ions present. ZIF-8 has large pores of up to 11.8 Å and a pore limiting diameter of 3.4 Å [2]. Moreover, ZIF-8 has a low hardness and elastic modulus, thus enabling it to have good elasticity among crystalline materials [37]. ZIF-7 has a hexagonal structure and is constructed by connecting zinc metal clusters with 1H-benzimidazole ligand to form a three-dimensional tetrahedral topological open framework [58]. The pore diameter of ZIF-7 ranges from 3.0 Å to 4.2 Å.

Besides that, copper (Cu)-based MOFs have also been highly researched on due to the potentially higher affinity for polar molecules compared with other metal organic frameworks because of the inclusion of unsaturated open metal sites [25]. For the MOF Cu_3_(BTC)_2_, it has a three-dimensional structure with small but highly electrostatic tetrahedron side pockets that have a pore size of 6 Å and cage sizes ranging from 10 Å to 12 Å. Furthermore, it also contains large but weak electrostatic square shaped channels connected by triangular windows of 3.5 Å diameter with the unsaturated Cu molecules being the center of the guest bindings [2,59]. On the other hand, Iron (Fe)-based MOFs are usually used to substitute Cu-based MOFs as iron is more reliable in toxic environments compared with copper [60]. Fe(BTC) MOF is comprised of iron trimeric octahedral clusters with similar vertex and removable H_2_O or OH ligands. Additionally, it contains two mesoporous cages with sizes of 25 Å and 29 Å and microporous windows of sizes 5.5 and 8.6 Å [60,61]. Table 1 shows the type of MOFs commonly reported in the literature and their structures.

## 4. Amine-Functionalized Metal-Organic Frameworks Fillers

Recently, MOFs are one of the most exciting materials for gas separation due to their distinct pore size and tunable functionalities. Incorporation of amine groups in MOFs was further developed because amine-functionalized MOFs exhibit higher affinity towards acidic gases such as CO_2_ and thus, are widely reported in the literature. Amine-functionalized MOFs contain several benefits which include: (i) highly selective gas adsorption properties and (ii) improved interaction between filler and polymer matrix for the formation of mixed matrix membranes. These amine-functionalized MOFs can be prepared mainly by three methods which are in situ synthesis, post-modification with amines, and physical mixing unfunctionalized MOFs and polyamines. Until now, grafting amines onto MOFs has been accomplished mainly by in situ synthesis with amine-containing ligands or through post-synthetic modification [69,70]. The various functionalization of MOFs are amine functions with ligands such as 2-amino-1,4-benzenedicarboxylate, 3-amino-1,2,4-triazole, and 4,4′,4′’-benzene-1,3,5-triyl-tribenzoic acid [70].

The inclusion of ligands containing additional functional moieties is not straightforward, because such groups may directly coordinate to the metal ions and therefore, inhibiting the MOF assembly depending on the reaction conditions used. An example of an amine-functionalized MOF is NH_2_-UiO-66. Due to the zirconium metal center being connected to the BDC-linker (benzene-1,4-dicarboxylate) with modifications with polar based functions such as with amino groups (-NH_2_), the NH_2_-UiO-66 is able to achieve a substantially higher chemical and thermal stability compared with the non-functionalized UiO-66 [35,47].

Furthermore, the supertetrahedral (ST) building blocks of MIL-101 (Cr) are generated by stiff terephthalate ligands. However, MIL-101’s existence of coordinatively unsaturated metal sites (CUSs) allows it to be used as a mild Lewis acid and, more importantly, to be post-functionalized through grafting of active species [71]. The functionalization of NH_2_-MIL-101(Al) can only be formed in specific synthesis conditions using solvothermal synthesis and using AlCl_3_ in DMF, where both the metal source and the solvent used play a key role, therefore achieving a higher thermal and chemical stability, high capacity and good regenerability [71]. Figure 2 shows the illustration of UiO-66, ZIF-8, and MIL-101 before and after amine functionalization, as well as the functionalization method. Generally, the pore size and surface area decrease after amine functionalization [72,73].

## 5. Amine-Functionalized Metal-Organic Frameworks Flat Sheet Mixed Matrix Membranes

Although MOFs based mixed matrix membrane (MMM) provides better separation performance, there are still a number of methods to be explored in order to enhance the MMMs’ performance such as the introduction of amine groups into the pore structure of MOFs which leads to the preferential adsorption of CO_2_ [17]. Apart from that, the presence of amine functional groups on the surface of the MOF crystals is considered beneficial to improve the compatibility between the dispersed filler material and the polymeric matrix, reducing interfacial defects between both phases which are known to be detrimental for the separation performance [79]. The advantages of the amine functionalization of MOFs in mixed matrix membrane for the separation of CO_2_/CH_4_ have been experimentally demonstrated in various studies reported in the literature [80,81]. MOFs are suitable as fillers for MMMs as most of their aperture sizes tend to be closer for the separation of CO_2_ which has a kinetic diameter of 3.3 Ȧ [47].

S. Meshkat et al. [38] incorporated MIL-53 and NH_2_-MIL-53 in poly(ether-b-amide) (PEBAX MH-1657) polymer for the fabrication of MMMs via a solution casting method for gas separation. Both the CO_2_ permeability and CO_2_/CH_4_ selectivity increased for MMM loaded with 10 wt% of filler. By increasing the filler loading up to 20 wt%, the permeability decreased significantly. They found that this result can be due to the formation of a rigid polymer layer around the MOF particles inside the matrix, the tortuosity introduced by the MOF, and MOF agglomeration leading to access restriction of gas molecules to a large portion of the porosity inside the matrix. Moreover, at filler loading of 10 wt%, the amine-functionalized NH_2_-MIL-53/PEBAX membrane demonstrated ideal selectivity of 20.5, which was lower than that of non-functionalized MIL-53/PEBAX membrane with an ideal selectivity of 23.3. This can be due to the presence of higher defects, formed in the membrane, possibly caused by the less compatibility between the amine-functionalized NH_2_-MIL-53 filler and PEBAX polymer. Guo et al. [82] tested NH_2_-MIL-125 (Ti)/PSf MMM at different operating pressures (3 and 10 bar). The CO_2_ permeability and CO_2_/CH_4_ selectivity increased as the loading of NH_2_-MIL-125 (Ti) increased at 3 bar. At pressure of 10 bar, the CO_2_ permeability increased as the loading of NH_2_-MIL-125 (Ti) increased but the CO_2_/CH_4_ selectivity showed a different trend. The increase in gas permeability was attributed to the extra pore network provided by the MOF particles for gas molecules while the reduction in selectivity at 10 bar indicated that the amine-functionalized membrane did not exhibit good resistance towards high pressure.

In another work, Waqas et al. [17] fabricated MIL-125/Matrimid and NH_2_-MIL-125/Matrimid for CO_2_ and CH_4_ gasses separation. They found that CO_2_ permeability increased as the filler loadings increased for both fillers, but the CO_2_/CH_4_ selectivity showed an opposite trend. Cross-sectional SEM images of the MMMs fabricated by Waqas et al. [17] are shown in Figure 3. Referring to Figure 3, the membrane displays little to no filler agglomeration for filler loadings of 5% and 15% which indicates that fillers and polymer have a good interaction. However, when the loading of fillers is increased to 30%, large filler agglomeration can be observed especially for NH_2_-MIL-125. Moreover, NH_2_-MIL-125/Matrimid MMMs exhibited a more significant increase in CO_2_ gas permeance compared with the MIL-125/Matrimid MMM by 85.18% while having similar CO_2_/CH_4_ gas pair selectivity at filler loading of 30%. The increase in gas permeation performance was possibly due to the good interaction between the amine-functionalized filler and the polymer because of the hydrogen bonding interaction between the -NH_2_ group at the filler surface and the polymer chains.

Meanwhile, Nik et al. [83] fabricated mixed matrix membranes by incorporating various MOFs including UiO-66 and MOF-199 with their amine-functionalized counterparts NH_2_-UiO-66 and NH_2_-MOF-199 into 6FDA-ODA polymer for the aim of determining the ligand functionalization effect (-NH_2_) on the adsorption properties and CO_2_/CH_4_ gas permeation and selectivity. Figure 4 shows the CO_2_ and CH_4_ adsorption isotherms of several types of MOFs at 35 °C based on the BET surface areas of the adsorbents. The adsorption isotherms shows that the amine-functionalized NH_2_-MOF-199 and NH_2_-UiO-66 have higher CO_2_ adsorption capacities compared with its non-functionalized counterparts. Cross-sectional SEM images of the UiO-66/6FDA-ODA and NH_2_-UiO-66/6FDA-ODA MMMs in Figure 5 shows that the amine-functionalized fillers were well-distributed in the 6FDA-ODA polymer matrix due to the presence of hydrogen bonding between -NH_2_ in the filler and carboxylic acid groups in the polymer chain. However, the interface between UiO-66 filler and the 6FDA–ODA polymer was found to have inferior quality. With the addition of the -NH_2_ group in NH_2_-UiO-66/6FDA–ODA, the filler/polymer interfacial interaction was shown to improve. For gas permeation performance, their results obtained for NH_2_-UiO-66/6FDA–ODA MMM showed a reduction in permeability by 5% and a massive 267% compared with the neat 6FDA–ODA and UiO-66/6FDA–ODA, respectively. Whereas, for the CO_2_/CH_4_ selectivity, the NH_2_-UiO-66/6FDA–ODA MMM demonstrated an increment by 17% and 18% compared with the neat 6FDA–ODA and UiO-66/6FDA–ODA, respectively.

Figure 6 summarizes the performance of several MMMs with different polymer-filler combinations into a Robeson plot. Table 2 summarizes the performances of the non-functionalized and amine-functionalized MOFs based mixed matrix membrane for CO_2_/CH_4_ separation. In most MMMs, amine-functionalized MMMs showed improved gas separation performance compared with MMM loaded with non-functionalized MOF.

## 6. Amine-Functionalized Metal-Organic Frameworks Hollow Fiber Mixed Matrix Membranes

Hollow fiber membranes (HFMs) are semipermeable cylindrical or capillary-shaped membranes with internal and exterior diameters of less than 0.25 mm and 1.00 mm, respectively. Hollow fiber configuration is more suitable for industrial gas separation due to their high surface to volume ratio and selectivity thus giving exceptional mass transfer properties for gas separations [86,87,88]. Hollow fiber mixed matrix membranes with a wide surface area and thin selective layers are recommended in most cases as the thickness of gas separation membranes is important [37,89]. To further enhance the performance of MMMs, incorporation of fillers into HFM configuration will form a hollow fiber mixed matrix membrane (HFMMM). HFMMMs have a significant advantage over conventional flat sheet membranes and hollow fiber membranes such as a better separation factor, mechanical strength, and thermal and chemical resistivity [90,91].

### 6.1. Hollow Fiber Mixed Matrix Membranes

Several advantages may be gained by incorporating high performance potential MMMs into a hollow fiber membrane configuration to generate HFMMMs. HFMMMs have a number of advantages over conventional flat sheet and hollow fiber membranes, including a higher separation factor, mechanical strength, and heat and chemical resistance [92]. Among the most popular methods currently utilized for the fabrication of HFMMMs which was first developed by Mahon at Dow Chemical in the 1960s is the solution spinning methods such as dry phase inversion spinning, wet phase inversion spinning, and dry-jet wet phase inversion spinning [93]. Figure 7 shows a spinning machine used for dry-jet wet phase inversion which is the most commonly used method to fabricate hollow fiber mixed matrix membranes for gas separation [94].

Sasikumar et al. [72] fabricated hollow fiber mixed matrix membranes (HFMMMs) which incorporated ZIF-8, S/ZIF-8, and amine-modified S/ZIF-8 (A@S/ZIF-8) of 0.5 wt% into polysulfone (PSf) matrix using a dry-wet phase inversion technique. Their research found that the addition of an amine group in the S/ZIF-8 MOF enhanced the interaction between the filler and PSf matrices, therefore improving the CO_2_/CH_4_ gas selectivity. The best performing HFMMM was achieved by the amine-functionalized A@S/ZIF-8 with a CO_2_ permeability of 41.15 GPU and CO_2_/CH_4_ selectivity of 22.25. The results show a selectivity increment of 61.46% compared with neat PSf hollow fiber membrane and 42.9% compared with ZIF-8/PSf HFMMM. They concluded that the facilitated CO_2_ transport with the amine group incorporated on the membrane results in increased membrane performance.

To date, there have not been sufficient works conducted on the use and comparison of HFMMMs before and after amine functionalization for gas separation despite the large potential it may have. Therefore, further efforts should be made focusing on this area for the investigation of amine-functionalized HFMMMs on gases separation.

### 6.2. Casting on PEBAX Thin Selective Layer

Thin layer composite membranes are formed by coating the outer layer of membrane surface with a selective layer and a highly permeable gutter layer [95]. PEBAX, which is a rubbery block copolymer containing Polyether oxide (PEO) segments, is used as the selective layer as it is reported to be high selective towards carbon dioxide [96]. The use of mixed matrix selective layer was first reported by Jia et al. in which they prepared a zeolite silicalite-1/PDMS selective layer on polyetherimide for O_2_ and N_2_ gases separation [97]. Thin layer composite membranes are preferred as they can manage large gas flux needs and are one of the most energy-efficient and industrially feasible CO_2_ capture possibilities [98].

Sutrisna et al. [37] incorporated UiO-66, UiO-66-NH_2_, UiO-66-(COOH)_2_, and ZIF-7 into a thin PEBAX selective layer on a PVDF support hollow fiber configuration as shown in Figure 8 with loadings from 30% for ZIF-7 and up to 80% for UiO-66. The choice of using a PEBAX/MOF selective layer instead of the conventional HFMMM fabrication was due to the PEBAX layering on the PVDF support provides a higher resistance towards plasticization. The CO_2_ and CH_4_ gas permeance and the CO_2_/CH_4_ selectivity was determined using pure gas. Hence, the best performing membrane was determined to be NH_2_-UiO-66/PEBAX with 50% loading giving a CO_2_ permeance of 320 GPU and CO_2_/CH_4_ gas pair selectivity of 23 at pressures of 2 bar. These results show an increase in 45% and 44% for the gas permeation and selectivity, respectively, compared with the non-amine functionalized UiO-66. This is possibly due to the addition of amine groups on the organic ligands which boosted UiO-66-NH_2_ selective CO_2_ absorption, resulting in greater CO_2_ solubility inside the PEBAX layer. The research also discovered that by adding nanoparticles into PEBAX polymer, the rigidity of the polymer increased due to the presence of more hydrogen bonds. The more rigid PEBAX polymer is preferred as it retains the molecular sieving capabilities. Table 3 shows the performances of non-functionalized and amine-functionalized MOFs based hollow fiber mixed matrix membrane for CO_2_/CH_4_ separation.

## 7. Challenges in Development of Amine-Functionalized Metal-Organic Frameworks (MOFs) Hollow Fiber Mixed Matrix Membranes

With the high rate of progress in the development of MOFs as fillers in HFMMM fabrication, most commercial polymers can be potentially superseded by HFMMMs due to them being a superior membrane-based gas separation technology with a larger surface area, and tunable functionalities including amine functionalities. While numerous studies and research have been conducted to overcome the initial challenges for developing HFMMMs, several challenges are still present when developing HFMMMs for the use of gas separation such as polymer-filler incompatibility, filler particle agglomeration, and plasticization which can hinder the separation performance of the membrane [9,32,101].

### 7.1. Polymer-Filler Interfacial Incompatibility

Compatibility between a filler with the polymer matrix is a key factor in the development of HFMMMs. Careful material selection is required for a compatible polymer-filler interaction [23,102]. Moreover, certain MMMs such as those based on zeolite type fillers tend to have poor adhesion between the polymer and filler which leads to void defects formed on the material interface causing gas molecules to non-selectively diffuse through the voids therefore reducing the separation performance of the membrane [103]. The causes of interfacial incompatibility are the dewetting of the polymer at the interface and differences in mechanical properties between components, particularly for glassy polymers. However, when rubbery polymers are utilized, the polymer phase can obstruct the porosity of MOFs through pore infiltration, therefore limiting the MOF’s molecular sieving capabilities [90,104]. Figure 9 shows the various cases HFMMMs may experience depending on the compatibility of the filler and the polymer matrix. Case 1 is an ideal condition in which the filler is completely integrated into the polymer matrix with no sign of incompatibility. In Case 2, the polymer matrix becomes rigid in the vicinity of the filler which leads in an increase in selectivity, but also in a decrease in permeability. Case 3 shows the formation of an interphase as a result of the particle’s incompatibility with the polymer matrix which leads to an increase in permeability and a decrease in selectivity as a result of the larger fractional free volume. Finally, in Case 4, polymer matrix penetrates the pores or empty spaces of the filler; therefore, both the permeability and the selectivity decrease significantly [105].

Various methods can be used to counteract the incompatibility and pore blockage challenges such as through careful material selection consideration between the polymer matrix and MOF filler since performance improvements are only present when the polymer-filler compatibility is achieved [106]. Besides that, the incorporation of amine-functionalized groups into organic MOF linkers increases compatibility due to the presence of more hydrogen bonding with the hydroxyl groups present in polymer interface [79,107].

### 7.2. Agglomeration of Fillers

MOF loading also presents an important factor to consider when developing MMMs as higher filler loading will cause the gas transport to be more based on the MOF’s transport mechanism which can potentially provide a larger separation performance. Although, a higher MOF loading can cause a severe issue in the form of filler agglomeration [32]. The formation of particle agglomeration in MMMs are undesirable because it will cause pinhole defects, which reduce the membrane gas performance by affecting the selectivity of the gases [108]. Therefore, this results in a trade-off between the MOF loading and the gas pair selectivity of the MMM. To increase the MOF loading tolerance, physical priming can be performed on MOF surfaces. Physical priming is the deposition of the polymer dope solution on the MOF surface covalently thus allowing adhesion between the polymer and filler to increase which consequently increases the polymer-filler compatibility [32]. Figure 10 shows the cross-sectional and outer surface SEM images of PEBAX^®^ 2533-NH_2_-UiO-66/PP thin film HFMMM with the presence of severe agglomerations found as the NH_2_-UiO-66 loading increases to 20.0 wt% and 50.0 wt% based on the works by Li et al. [73]. They determined that with the severe agglomeration led to a decrease in CO_2_/N_2_ ideal selectivity and a 9% increase in CO_2_ permeance compared with the neat PEBAX^®^ 2533 membranes.

Therefore, there are also a number of aspects by which particle agglomeration may occur which includes: (1) larger MOF filler size, (2) MOF interparticle interaction whereby the MOF to MOF interaction is stronger than MOFs to the polymer, and (3) fabrication conditions [109]. Additionally, the dispersion of the fillers into the polymer is essential to promote good polymer-filler interactions. The dispersion of the fillers however is affected by the MOF’s size, geometry, and surface chemistry [32]. Therefore, by adjusting these parameters, an optimized MOF can be produced for MMM development. The size and geometry of the MOF can be altered by manipulating the synthesize conditions such as NH_2_-MIL-101(Al) which can be formed under very specific synthesis conditions, where both the metal source and the solvent used perform a key role [71], or by utilizing nanosized MOF fillers. While the surface chemistry of MOFs can be tuned using modification such as amine functionalization [36].

### 7.3. Plasticization

The organic polymer matrix acts as the continuous phase in HFMMMs and they too can suffer from certain limitations such as plasticization. The incorporation of MOF fillers into the polymer matrix does reduce the effects of plasticization as the presence of sufficient filler loading is able to limit the polymer chain mobility. This potentially allows certain HFMMM with strong polymer-filler interactions to be able to withstand plasticization effects at high pressures of up to 20 bar [25]. However, due to the large percent of polymer present compared with MOFs, plasticization remains a challenge when fabricating HFMMMs with organic polymers. Plasticization can be defined as a phenomenon that arises when the concentration of a strong sorbing gas such as CO_2_ in a polymer matrix increases, causing the polymer to swell and raise the diffusion coefficients of all gases present, resulting in a drop in selectivity [110]. When plasticization occurs, usually at high pressures, the gas permeability for all gases increases significantly.

Through the introduction of amine-functionalized MOFs into the polymer matrix, plasticization effects may be reduced. In the works conducted by Rajati et al. [111], they found that by increasing the loading of NH_2_-MIL-101 (Cr) up to 7 wt% led to a higher resistance for the Matrimid polymer to plasticize, from 12 bar in the neat Matrimid to 26 bar. This can be explained by the increase in resistance in the Matrimid chain mobility with the introduction of NH_2_-MIL-101 in the polymer matrix as the amine functional groups form strong hydrogen bonds that make the polymer more rigid and resistant to plasticization [111].

## 8. Conclusions and Future Directions

Mixed matrix membranes (MMMs) in hollow fiber configuration show great potential application in gas separation; however, the significant problems such as low compatibility and poor adhesion between the inorganic and polymer materials resulted in the creation of imperfect void structures and pinholes and which deteriorates the properties of the membrane. As a result, metal-organic framework (MOF)-based MMMs have gained attention in the past few years. Although MOF-based MMMs provide better gas separation performance compared with the other fillers, new methods to further enhance the gas separation performances are required due to certain limitations of the MOFs. Therefore, introducing amino functional groups on the surface of the MOF crystals is considered beneficial to improve the interaction between polymer matrices and the filler. This will prevent the formation of voids at the polymer-filler interface.

A comprehensive review reveals that amine-functionalized MOFs based MMMs for gas separation evidently showed that research in this field has grown intensely over the past few years due to its high potential in gas separation, but multiple challenges faced by the membranes hinder the possibly higher performance. By incorporating amine-functionalized MOFs into polymer matrix, it demonstrated advantages in gases separation. However, the compatibility between amine-functionalized MOFs and polymer as well as against reproducibility and stability of membrane performance are still needed to be addressed. Next, the parameters of fabrication also should be concerned in order to reduce the potential of agglomeration which can affect the performance of membrane. Besides that, hollow fiber mixed matrix membranes (HFMMMs) have a higher potential to surpass many of the shortcomings faced by conventional flat sheet MMMs due to its higher surface to volume ratio and better industrial applicability, yet in the field of CO_2_ gas separation, it is still relatively new and rarely explored. In the future, further research on amine-functionalized MOFs based MMMs and HFMMMs should be explored in order to fabricate the resultant membranes with minimal defects to provide higher performance in terms of permeability and selectivity under various operating conditions, particularly high pressure, before they are scaled up for industrial applications.

## Figures and Tables

**Figure 1 polymers-14-01408-f001:**
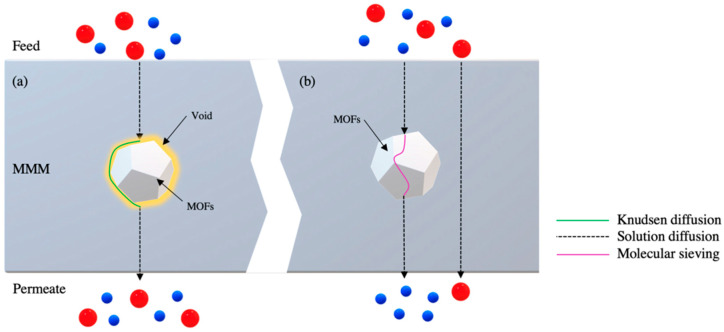
Transport mechanisms for gas transport through mixed matrix membrane with; (**a**) interfacial defect, (**b**) defect free layer [26].

**Figure 2 polymers-14-01408-f002:**
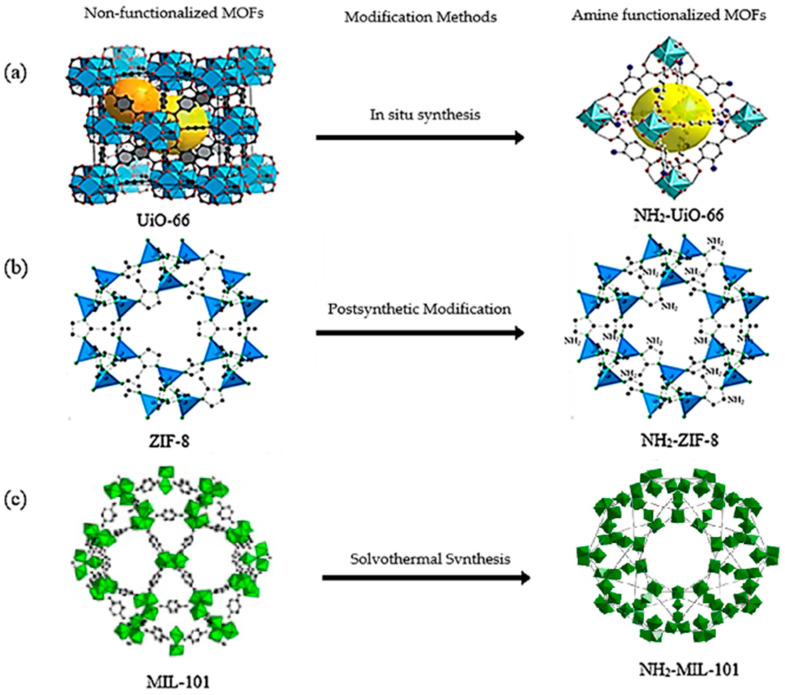
Schematic diagram of MOFs: (**a**) UiO-66 and NH_2_-UiO-66, (**b**) ZIF-8 and NH_2_-ZIF-8, (**c**) MIL-101 and NH_2_-MIL-101 [74,75,76,77,78].

**Figure 3 polymers-14-01408-f003:**
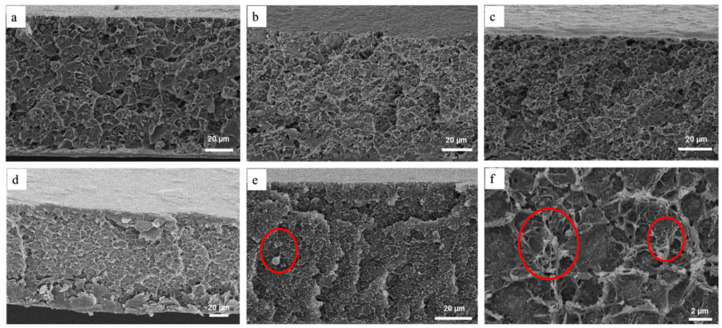
Cross-sectional SEM images of Matrimid with different filler loadings: (**a**–**c**) MIL-125 loading of 5%, 15%, and 30%, respectively, and (**d**–**f**) NH_2_-MIL-125 loadings of 5%, 15%, and 30%, respectively [17].

**Figure 4 polymers-14-01408-f004:**
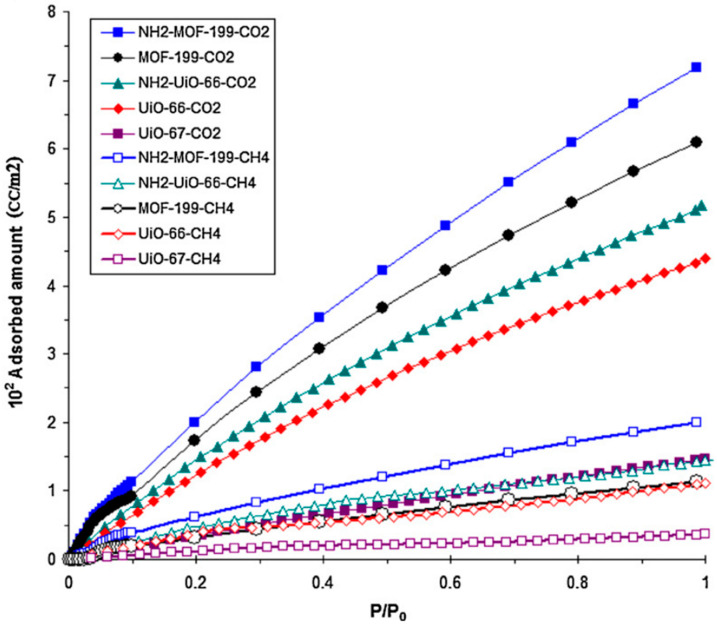
CO_2_ and CH_4_ adsorption isotherms of NH_2_-MOF-199, NH_2_-UiO-66, MOF-199, UiO-66, and UiO-67 reported by Nik et al. [83].

**Figure 5 polymers-14-01408-f005:**
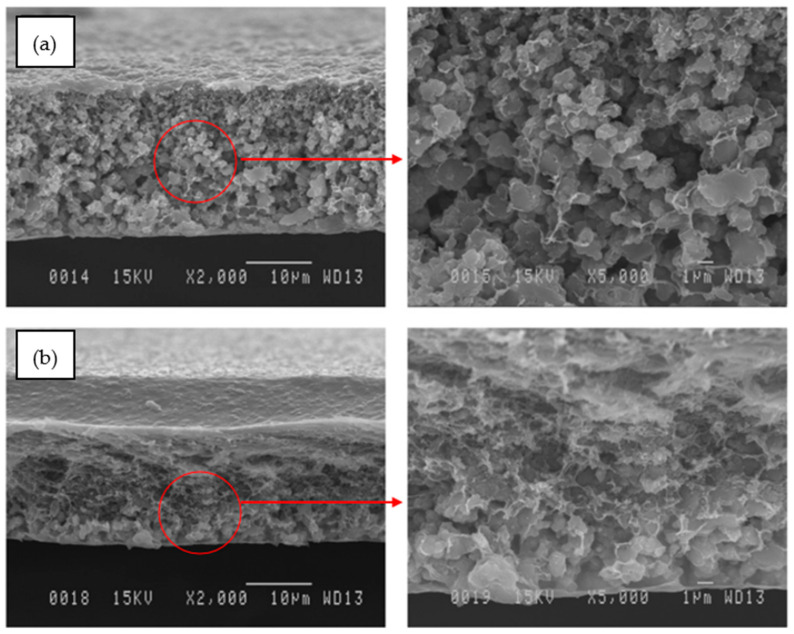
Cross-sectional SEM images of (**a**) UiO-66/6FDA-ODA and (**b**) NH_2_-UiO-66/6FDA-ODA [83].

**Figure 6 polymers-14-01408-f006:**
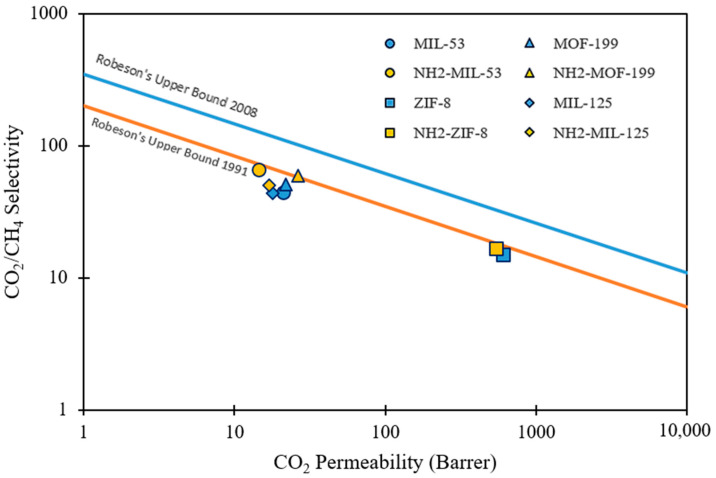
The Robeson plot of CO_2_/CH_4_ gas pair selectivity against CO_2_ permeability for MMMs; MIL-53/6FDA-ODA and NH_2_-MIL-53/6FDA-ODA [13], ZIF-8/6FDA-durene and NH_2_-ZIF-8/6FDA-durene [81], MOF-199/6FDA-ODA and NH_2_-MOF-199/6FDA-ODA [83], and MIL-125/Matrimid 9725 and NH_2_-MIL-125/Matrimid 9725 [17].

**Figure 7 polymers-14-01408-f007:**
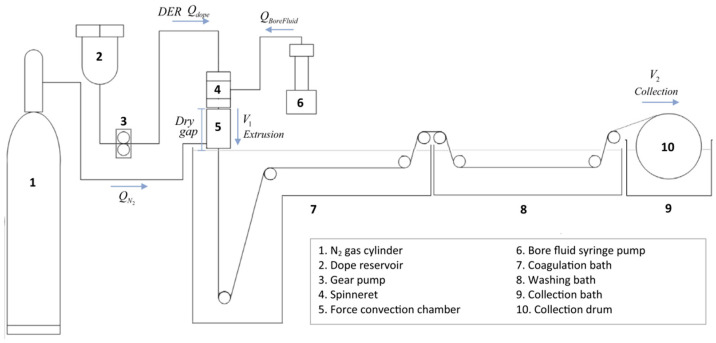
Schematic diagram for the spinning machine setup for the fabrication of hollow fiber mixed matrix membranes for gas separation [94].

**Figure 8 polymers-14-01408-f008:**
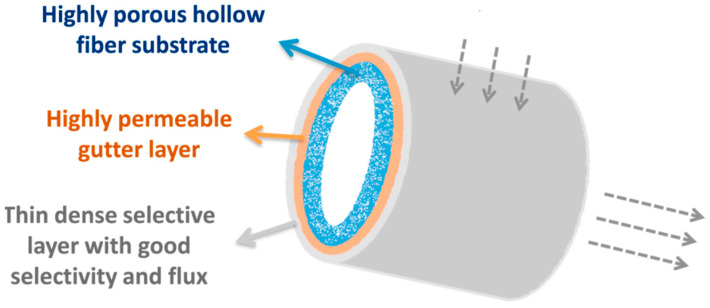
PEBAX thin selective layer [88].

**Figure 9 polymers-14-01408-f009:**
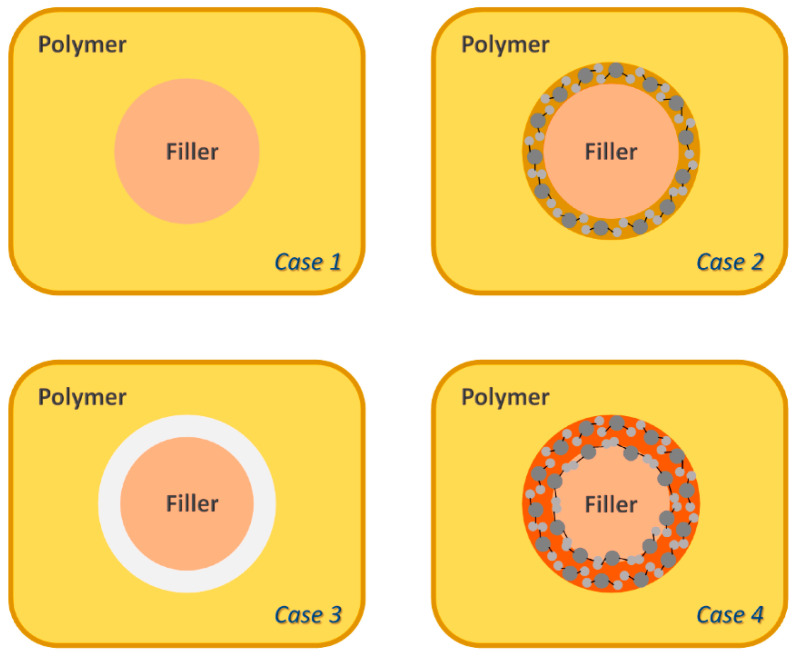
Schematic diagram on the types of cases for the polymer-filler interactions HFMMMs [105].

**Figure 10 polymers-14-01408-f010:**
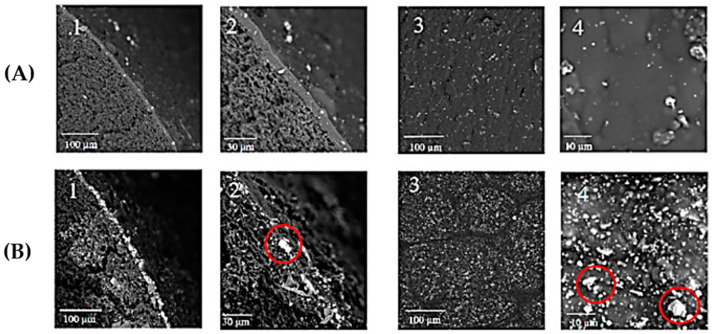
SEM images of cross-section (1 and 2) and outer surface (3 and 4) of PEBAX^®^ 2533-UiO-66NH_2_/PP thin film HFMMM-(**A**) 20 wt% NH_2_-UiO-66, and (**B**) 50 wt% NH_2_-UiO-66 [73].

**Table 1 polymers-14-01408-t001:** Illustration of MOF structures.

Metal-Organic Framework	Pore Size	MOF Structure	Reference
UiO-66	6.2 Å	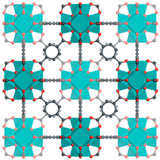	Reprinted with permission from Refs. [32,62]
UiO-67	8.0 Å	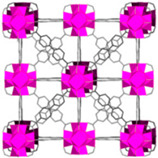	[51,63]
MIL-53 (Al)	8.5 Å	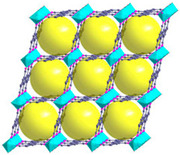	[38,64]
MIL-101 (Cr)	12 Å	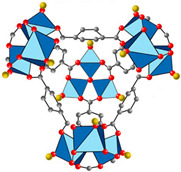	[32,65]
MIL-125 (Ti)	6 Å	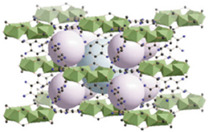	[63]
ZIF-4	2.1 Å	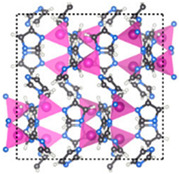	[66]
ZIF-7	3.0–4.2 Å	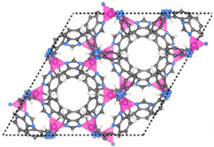	[58,66]
ZIF-8	3.4–11.8 Å	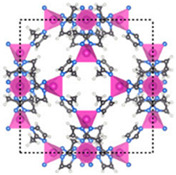	[2,66]
Cu_3_(BTC)_2_	6 Å		[67,68]
Fe-BTC	5.5–8.6 Å		[60,61]

**Table 2 polymers-14-01408-t002:** Summary of the performances of amine and non-amine functionalized MOFs based mixed matrix membrane for CO_2_/CH_4_ separation.

Amine-Functionalized MOFs MMM		Operating Parameters		Performance		Reference
Filler	Polymer	Particle Loading (%)	Temperature (K)/Pressure (Bar)	CO_2_ Permeability (Barrer)	CO_2_/CH_4_ Selectivity	
ZIF-8	6FDA-durene	5.0	298/3.5	693.5	16.5	[41]
APTMS-ZIF-8		0.5		649.6	17.4	[57]
AAPTMS-ZIF-8		0.5		825.1	26.2	
AEPTMS-ZIF-8		0.5		713.8	27.9	
APTMS-ZIF-8		1.0		518.3	13.5	
AAPTMS-ZIF-8		1.0		582.5	18.2	
AEPTMS-ZIF-8		1.0		561.6	15.0	
MIL-53	PEBAX MH-1657	5.0	311.15/10.0	127.4	18.7	[38]
		10.0		129.0	23.3	
		15.0		105.4	17.4	
		20.0		95.7	17.2	
NH_2_-MIL-53		5.0		134.6	19.1	
		10.0		149.1	20.5	
		15.0		125.6	20.6	
		20.0		63.3	17.3	
UiO-66	6FDA-DAM	14.0	308/3.0	1700.0	31.0	[84]
NH_2_-UiO-66		0.0		997.0	29.2	
		16.0		1223.0	30.0	
NH_2_-MIL-125 (Ti)	6FDA-durene	0.0	298/3.5	510.3	8.6	[80]
		1.0		922.8	23.0	
		3.0		930.0	27.0	
		5.0		1020.0	29.5	
		7.0		1115.7	37.5	
		9.0		961.8	31.0	
NH_2_-ZIF-8	PSF	0.0	300/4.0	59.0	18.0	[85]
		0.5		21.2	34.0	
ZIF-8	6FDA-durene	1.0	308/5.0	600.0	15.0	[81]
ZIF-8-AAPTMS		1.0		540.0	16.8	
ZIF-8-AEPTMS		1.0		520.0	11.5	
NH_2_-MIL-125 (Ti)	PSf	10.0	303/3.0	18.5	28.3	[82]
		20.0		29.3	29.5	
		30.0		40.0	29.2	
		10.0	303/10.0	15.0	28.5	
		20.0		22.8	29.5	
		30.0		36.8	5.7	
-	6FDA-ODA	-	308/10.0	14.4	44.1	[83]
UiO-66		25.0		50.4	46.1	
NH_2_-UiO-66				13.7	51.6	
MOF-199				21.8	51.2	
NH_2_-MOF-199				26.6	59.6	
MIL-125	Matrimid 9725	15.0	308/9.0	18.0	44.0	[17]
		30.0		27.0	37.0	
NH_2_-MIL-125		15.0		17.0	50.0	
		30.0		50.0	37.0	
MIL-53	6FDA-ODA	25.0	308/10.3	21.0	44.5	[13]
NH_2_-MIL-53		25.0		14.5	66.0	

**Table 3 polymers-14-01408-t003:** Summary of non-functionalized and amine-functionalized MOFs based hollow fiber mixed matrix membrane for CO_2_/CH_4_ separation.

Amine-Functionalized HFMMM		Operating Parameter		Performance		Reference
Filler	Polymer	Loading (%)	Temperature (K)/Pressure (Bar)	CO_2_ Permeability (GPU)	CO_2_/CH_4_ Selectivity	
UiO-66 in PEBAX	PVDF	50.0	298/2	220.0	16.0	[37]
NH_2_-UiO-66 in PEBAX				320.0	23.0	
NH_2_-MIL-53	CA	15.0	303/3	2.9	11.8	[99]
NH_2_-MIL-53	CA	15.0	298/3	6.7	16.0	[100]
ZIF-8	PSf	0.5	303/4	31.0	15.6	[72]
A@S/ZIF-8				41.2	22.3	

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
