# Peer review of "Development of Amine-Functionalized Metal-Organic Frameworks Hollow Fiber Mixed Matrix Membranes for CO2 and CH4 Separation: A Review"

_polymers, 2022, doi:10.3390/polym14071408_

Round 1
Reviewer 1 Report
It is a hot topic to design and prepare high CO2-selective membranes. This review tried to give a whole view of amine-functionalized MOFs mixed matrix membranes. It should revised before the acceptance.
(1) The authors emphasized CO2/CH4 separation in the title. To my best knowledge, natural gas upgrading was generally operated under high-pressure. However, only few references regarding high pressure separation were involved. On the other hand, the CO2/N2 separation using MMMs was also very hot, which might be involved. Therefore, the title and content should be reconsidered.
(2) In my opinion, the principle and mechanism should be reviewed first. Because that would be the criterion for the filler choose for optimized MMMs preparation.
(3) Table 2 is not easy for reading. Please reorganize the table. Meanwhile, some membrane performance was improved with amine-functioned MOFs while the others were not. Please review the detailed explanation.
(4) Robesen upper bound can well express the state-of-the-art membrane performance. Maybe the author can consider to summarize the membrane performance in this format.
(5) The MMM is beneficial to anti-plasticization and anti-aging. However, this kind of information was really short.
(6) The authors summarized different fillers. However, the ammine content in polymer was different. Is that a major factor for MMM design?
(7) Table 1, the MOF structure should be uniform. Such as, UiO-66 only give the metal nodes and UiO-67 illustrates the pore structure, while ZIFs only give the chemical bonds....
(8) Some spelling and concept should double checked. such as line 52-53, line 63-67, line 85, line 161, line 163-164.
Reviewer 2 Report
In this review, the authors discussed the amine-functionalized metal-organic framework based mixed matrix membranes and hollow fiber mixed matrix membranes. This topic is interesting. I would recommend the acceptance of the manuscript after the following revision:
- Metal-organic has a "-" in between, please correct it.
- The introduction section needs some reorganization. I suggest the authors to talk about MMM and some other commonly used inorganic fillers in paragraph 2, and then introduce the use of MOF in MMM in paragraph 3.
- The definition of MOF is introduced twice in both lines 53-54 and 85-86, I suggest to delete one.
- More general information about MOF should be discussed in either section 1 or section 2. For example, one or two sentences about the wide application of MOFs and the current MOF synthesis allows the formation of MOF structures in 0D-3D dimensions. Also, one or two sentences about the fact that pure MOF films are fragile and brittle, and MOF based MMM is an alternative approach towards pure MOF membranes. The following papers on MOF synthesis (Communications Chemistry 4.1 (2021): 1-10; Chemistry–A European Journal 26.61 (2020): 13788-13791) and MOF based MMM (Chemical Communications 52.100 (2016): 14376-14379.; Nature Reviews Materials 1.12 (2016): 1-17.) can be cited in their respective context.
- I suggest the authors to include more information in Table 1, for example including the reported pore size, surface area for each MOF in the table.
- Figure 1 is too simple. In addition to just include the structures of MOFs before and after amine functionalization, it's better to also give detailed information about which methodology was used to functionalize the MOF, and how the pose size and surface area of the MOFs changed after amine functionalization.
- I suggest the authors to include one figure with a typical CO2 adsorption isotherms for MOF base MMM.
Round 2
Reviewer 1 Report
the authors revised well the manuscript and no further comments.